# Point-of-Care Ultrasonography as an Extension of the Physical Examination for Abdominal Pain in the Emergency Department: The Diagnosis of Small-Bowel Volvulus as a Rare Complication after Changing the Feeding Jejunostomy Tube

**DOI:** 10.3390/diagnostics12051153

**Published:** 2022-05-06

**Authors:** Tse-Chyuan Wong, Rhu-Chia Tan, Jian-Xun Lu, Tzu-Heng Cheng, Wei-Jun Lin, Te-Fa Chiu, Shih-Hao Wu

**Affiliations:** 1Department of Emergency Medicine, China Medical University Hospital, Taichung 404, Taiwan; d28535@mail.cmuh.org.tw (T.-C.W.); rhuchiachen@gmail.com (R.-C.T.); bbgun1208@gmail.com (W.-J.L.); tefachiu@gmail.com (T.-F.C.); 2College of Medicine, China Medical University, Taichung 404, Taiwan; 3Department of Emergency Medicine, Chang Gung Memorial Hospital and Chang Gung University, Taoyuan 333, Taiwan; djf4siuol@hotmail.com (J.-X.L.); b9502086@cgmh.org.tw (T.-H.C.); 4Department of Emergency Medicine, New Taipei Municipal Tucheng Hospital, New Taipei City 236, Taiwan

**Keywords:** complication, volvulus, whirlpool sign, point-of-care ultrasound, computed tomography/CT

## Abstract

Point-of-care ultrasonography (POCUS) has become the most popular modality of testing for physicians in recent years and is used for improving the quality of care and increasing patient safety. However, POCUS is not always acceptable to all physicians. To address the benefits and importance of POCUS, numerous studies have examined the use of POCUS in clinical practice and even medical education. This article aims to highlight the effects of POCUS as an extension of the physical examination, and we present a case to address the reasons it should be performed. For a man experiencing abdominal pain immediately after his feeding jejunostomy tube was changed, there was high suspicion of small-bowel volvulus after a “whirlpool sign” was observed during the POCUS, whereby mesenteric vessels presented in a whirling or spiral shape. This impression was subsequently confirmed by computed tomography. Small-bowel volvulus is a rare complication of changing a feeding jejunostomy tube. The images submitted here add to the sparse evidence from the literature on the use of POCUS as an extension of the physical examination for evaluating abdominal pain. POCUS can be used after taking the patient’s history and conducting a physical examination. The observation of a whirlpool sign may indicate the presence of a volvulus that is life-threatening.

**Figure 1 diagnostics-12-01153-f001:**
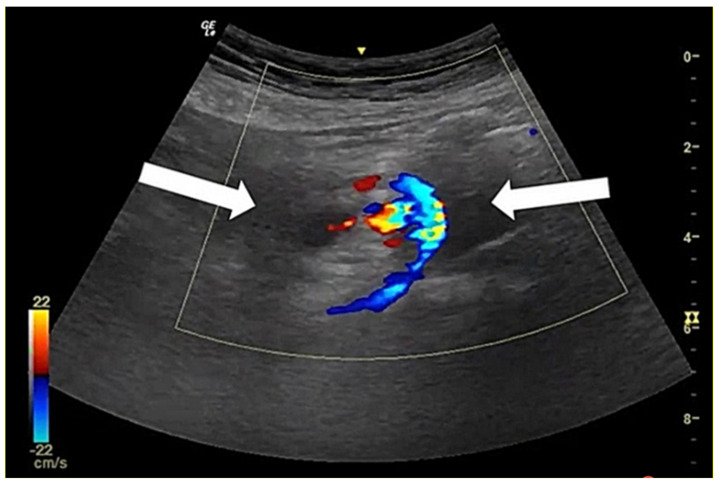
Point-of-care ultrasonography image demonstrating the “whirlpool sign” (arrows) over the mesenteric vessels as indicated by their presentation in a spiral shape. Abdominal pain is the most common complaint in the emergency department (ED) [1,2]. Acute abdominal pain can be caused by a spectrum of conditions ranging from benign and self-limited diseases to surgical emergencies. Abdominal pain is caused by a variety of gastrointestinal and non-gastrointestinal disorders. Some frequently missed conditions include biliary pathology, appendicitis [3], diverticulitis, urogenital pathology, and even vascular emergency [1]. Therefore, the rapid and early detection of urgent conditions is crucial for managing patients efficiently. The combination of clinical and laboratory evaluation cannot reliably predict or exclude urgent conditions and can result in unnecessary or delayed interventions. Further diagnostic imaging such as point-of-care ultrasonography (POCUS) can help in the early identification of the cause of abdominal pain [4,5] and increase the certainty of the diagnosis [6]. The American College of Emergency Physicians (ACEP) defines clinical ultrasonography as a diagnostic modality that provides clinically significant data that are not obtainable by inspection, palpation, auscultation, or other components of the physical examination [7]. POCUS performed and interpreted by physicians at the bedside has grown rapidly in recent decades [8], as current ultrasound equipment has become less expensive, higher quality and more compact. The use of bedside ultrasonography (or insonation), incorporated with traditional inspection, palpation, percussion, and auscultation have become the five pillars of bedside clinical medicine [9]. As an extension of the physical examination [10], a more generalized concept of an “ultrasound stethoscope” is used to provide directed clinical assessments [11,12]. It could provide immediate, real-time dynamic images that are correlated with a patient’s clinical condition and are easily repeatable. The main goal of POCUS in ED is to rapidly rule in or rule out a dangerous diagnosis and to solve clinical problems, such as evaluating the cause of shock [13]; shortness of breath [14]; chest pain; fever; and muscular, skeleton, or soft tissue swelling and pain [15,16] before laboratory tests. It can also help to guide the procedure [17,18] and allow checking for complications to promote patient safety [19]. Moreover, it can be used as a first-line tool for the evaluation of abdominal pain after taking the patient’s history and conducting a physical examination [4,5,20,21]. The use of ultrasonography can enable the detection of urgent conditions in patients with acute abdominal pain and can decrease unnecessary radiation exposure [6], decrease the need for further examinations, and decrease the frequency with which patients are admitted [22]. Nevertheless, ultrasonography is an extremely operator-dependent modality of testing. Acquiring the skills to properly manipulate and interpret images for safe integration with clinical work takes years to master [20,23]. There are many factors that can affect the quality of POCUS and the decision making involved, such as patient factors, the operator’s skills in image acquisition and interpretation, and machine quality [11]. It is important to be able to identify and differentiate artifacts and false positives and negatives to overcome the pitfalls of POCUS [24,25]. Therefore, POCUS education has been integrated into postgraduate-year training, and emergency resident training is important. It will result in physicians having more self-confidence to overcome the barriers for implementing POCUS in clinical practice. Here, we present an unusual case of abdominal pain after jejunostomy tube insertion with the use of POCUS for diagnosis. A 51-year-old male with a history of esophageal cancer presented to the emergency department due to a jejunostomy tube that had been accidentally dislodged without causing abdominal pain and hemodynamically stable. A few minutes after the introduction of a new tube, the patient complained of diffuse abdominal pain. Upon examination, his body temperature was 36.6 °C, his pulse rate was 79 beats per minute, his blood pressure was 90/45 mmHg, and his respiratory rate was 18 breaths per minute. The physical examination revealed a soft abdomen without rebounding tenderness or muscle guarding. The jejunostomy tube functioned well, and laboratory tests were unremarkable. The source of abdominal pain was considered benign. However, analgesic agents could not relieve abdominal pain. We performed POCUS for persistent abdominal pain to rule out bowel perforation or other dangerous etiology such as vascular emergencies, and it demonstrated mesenteric vessels presenting in a whirlpool sign, namely, in a whirling or spiral shape (Figure 1, Appendix A). We tried to remove the tube; however, it could not be moved and induced progressive pain. It prompted the use of computed tomography (CT) (Figure 2, Appendix A). This confirmed the presence of a small-bowel volvulus with a whirlpool sign over the jejunal branches of the superior mesenteric vessels. A surgeon was consulted, and this patient underwent surgical correction of the volvulus, without subsequent complications.

**Figure 2 diagnostics-12-01153-f002:**
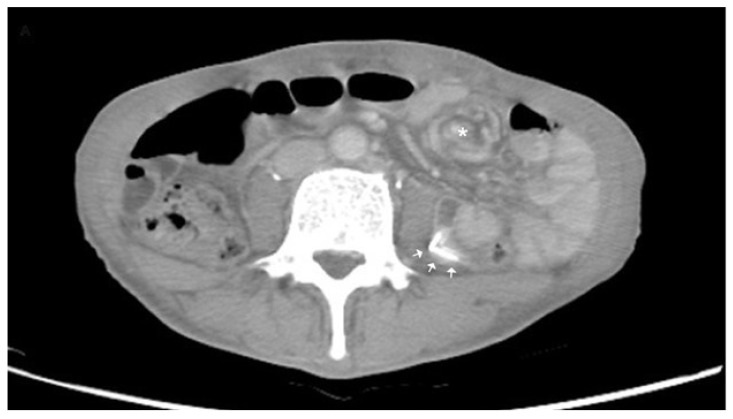
Abdominal computed tomography demonstrating the “whirlpool sign” over the jejunal branches of the superior mesenteric vessels (asterixis) with the feeding jejunostomy tube in place (arrows). In fact, if we conducted POCUS immediately after the physical examination, as an extension of the physical examination, we could obtain the final diagnosis faster without waiting for laboratory tests and symptom relief. This is where the real usefulness of POCUS lies—in speeding up diagnosis and management. Owing to how crowded and busy the emergency department is, many physicians order laboratory studies just after taking patient history and conducting physical examinations without POCUS. If the patient does not have a peritoneal sign and laboratory tests are not abnormal, physicians will screen for abdominal pain. If any of the above changes or abdominal pain persists or worsens, the physicians will arrange an abdominal CT for further confirmation. Abdominal CT has been shown to reduce early return visits [26]. However, if a patient suffers from a vascular emergency such as superior mesenteric artery dissection [27] or abdominal aortic aneurysm [28], “waiting for the laboratory tests” will result in a disaster, due to interventions being delayed. Moreover, abdominal pain is the leading presenting symptom, which accounts for 31% of the symptoms among patients who spend >4 h in the ED [29]. If we perform POCUS immediately after taking the patient’s history and conducting a physical examination, we will obtain the impression earlier, and it will help us to obtain the correct disposition faster. Moreover, it will help to create more order in the emergency department. It may help to relieve overcrowding in the ED and decrease inpatient mortality, the length of stay, and the costs for the admitted patients [30]. However, there is still no direct evidence for this. In the literature, there is only evidence that POCUS could reduce the disposition time in patients with dyspnea [31] and deep vein thrombosis [32]. Feeding jejunostomy is a common surgical procedure for enteral nutrition. However, complications that require re-exploration and that can be life-threatening may develop. Common complications include tube dislocation, abdominal wall or intra-abdominal infection, gastrointestinal symptoms, bowel necrosis, pneumatosis intestinalis [33], fluid and electrolyte imbalances [34], enteral migration [35], and intussusception [36]. Small-bowel volvulus, which refers to the torsion of the alimentary tract, is a rare complication of changing the jejunostomy tube [33,37]. A patient with volvulus may present with abdominal pain, abdominal distension, constipation, nausea, or vomiting. The characteristics of whirlpool signs (mesenteric vessels that have a whirling or spiral shape) can be detected via POCUS [38] or CT. Although abdominal CT is considered the optimal tool for diagnosis [39], POCUS can detect specific and dynamic signs of small-bowel volvulus [38] with no radiation, contrast exposure, lesser expenses, and higher availability. Patients who present with small-bowel volvulus should obtain immediate surgical consultations. If left untreated, it may eventually lead to catastrophic bowel ischemia, necrosis, and perforation [40]. Our case illustrates that abdominal pain immediately after jejunostomy tube insertion is a sign of life-threatening iatrogenic small-bowel volvulus with a characteristic “whirlpool sign”, which may be detected by POCUS. The pain may be mimicked by benign colic or traction pain caused by the tube. If we performed POCUS as an extension of the physical examination, we would obtain the final diagnosis faster. In addition to the utilization of POCUS in ED patients with abdominal pain, this review identified whether POCUS could reduce the disposition time, length of stay in the ED, and number of return visits for patients with abdominal pain in the ED, as an area for potential policy research and future exploration.

## Data Availability

No new data were created or analyzed in this review. Data sharing is not applicable for this article.

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
