# Peer review of "Point-of-Care Ultrasonography as an Extension of the Physical Examination for Abdominal Pain in the Emergency Department: The Diagnosis of Small-Bowel Volvulus as a Rare Complication after Changing the Feeding Jejunostomy Tube"

_diagnostics, 2022, doi:10.3390/diagnostics12051153_

Round 1
Reviewer 1 Report
The authors extended the introduction and the discussion part. Supplementary references were added from the first submission. Thus, I recommend publication.
Author Response
Thank you for your nice comments on our article.
Reviewer 2 Report
My congratulations to authors for their revised paper.
I do not recommend the publication of the manuscript in its present form because the authors did not :
1 - Improve the presentation of the case, following suggestions
2 - No novelty based on the authors' experience was included in case presentation or discussion.
Author Response
Thank you for your nice comments on our article. Even there is no novelty based on the our experience was included in case presentation or discussion. However, there are still many physicians who perform clinical practice without POCUS. This article aims to highlight the effects of POCUS as an extension of the physical examination, and we present a case to address the reasons it should be performed. To promote POCUS in emergency medicine and general medicine, it may help to relieve overcrowding in the ED and improve care quality and patient safety.
Round 2
Reviewer 2 Report
My congratulations to the authors for their improvement of the manuscript.
Considering that the revised manuscript is the best they can do, I recommend the publication of this article without any priority.
This manuscript is a resubmission of an earlier submission. The following is a list of the peer review reports and author responses from that submission.
Round 1
Reviewer 1 Report
My congratulation to the authors for their interesting case. Unfortunately, this manuscript is unacceptable for publication due to the following aspects:
1 - A case report to be published must be extremely rare, or bring up original novelty in propedeutics or treatment, or have an appropriate discussion based on an extense literature with authors' comments based on their professional experience.
None of these request can be found in this manuscript.
2 - An introduction must be included with background and justification of the importance to this case be presented.
3 - This case has an extremely poor presentation.
4 - No relevand discussion could be found.
I suggest the authors to look for a mentor to teach them how to write a scientific paper and submit another paper in the future.
Reviewer 2 Report
The manuscript represents a small case report that can be developed.
The introduction part about small bowel volvulus can be extended and more references added.
I recommend authors to extend the discussion part as well. They could include the importance of point-of-care ultrasound and its applications, for abdominal diagnosis and even for chest. (Cite: Constantin V, Carap AC, Zaharia L, Bobic S, Ciudin A, Brătilă E, Vlădăreanu V, Socea B. High correlation of lung ultrasound and chest X-ray after tube drainage in patients with primary spontaneous pneumothorax: can we omit X-rays for tube management? Eur Surg, 2015, 47(4): 175-180, DOI 10.1007/s10353-015-0333-9.)
Overall, more references are needed.